# Comparative 4-year risk and type of hospital admission among homeless and housed emergency department attendees: longitudinal study of hospital records in England 2013–2018

Charlie Moss [ID],[1] Matt Sutton,[1,2] Sudeh Cheraghi-Sohi,[3] Caroline Sanders,[3] Thomas Allen [ID] [4,5]

► Prepublication history and supplemental material for this paper is available online. To view these files, please visit the journal online (http://dx.doi.org/10.1136/bmjopen-2021-049811).

For numbered affiliations see end of article.

**Correspondence to**
Charlie Moss;
charlie.moss@manchester.ac.uk

## ABSTRACT

**Objectives** People experiencing homelessness are frequent users of secondary care. Currently, there is no study of potentially preventable admissions for homeless patients in England. We aim to estimate the number of potentially preventable hospital admissions for homeless patients and compare to housed patients with similar characteristics.

**Design** Retrospective matched cohort study.

**Setting** Hospitals in England.

**Participants** 16 161 homeless patients and 74 780 housed patients aged 16–75 years who attended an emergency department (ED) in England in 2013/2014, matched on the basis of age, sex, ED attended and primary diagnosis.

**Primary and secondary outcome measures** Annual counts of admissions, emergency admissions, ambulatory care-sensitive (ACS) emergency admissions, acute ACS emergency admissions and chronic ACS emergency admissions over the following 4 years (2014/2015–2017/2018). We additionally compare the prevalence of specific ACS conditions for homeless and housed patients.

**Results** Mean admissions per 1000 patients per year were 470 for homeless patients and 230 for housed patients. Adjusted for confounders, annual admissions were 1.79 times higher (incident rate ratio (IRR)=1.79; 95% CI 1.69 to 1.90), emergency admissions 2.08 times higher (IRR=2.08; 95% CI 1.95 to 2.21) and ACS admissions 1.65 times higher (IRR=1.65; 95% CI 1.51 to 1.80), compared with housed patients. The effect was greater for acute (IRR=1.78; 95% CI 1.64 to 1.93) than chronic (IRR=1.45; 95% CI 1.27 to 1.66) ACS conditions. ACS conditions that were relatively more common for homeless patients were cellulitis, convulsions/epilepsy and chronic angina.

**Conclusions** Homeless patients use hospital services at higher rates than housed patients, particularly emergency admissions. ACS admissions of homeless patients are higher which suggests some admissions may be potentially preventable with improved access to primary care. However, these admissions comprise a small share of total admissions.

## Strengths and limitations of this study

► Large sample of 90 941 patients.
► Comparative study design.
► Use of linked mortality data to avoid immortality bias.
► 'No fixed abode' flag used to identify patients who are homeless is not a perfect indicator of homelessness.
► Use of emergency department attendance data to identify study participants may affect generalisability to less-regular emergency department attendees.

## BACKGROUND

Homelessness remains a problem in England, as in many developed countries.[1 2] The homelessness charity, Shelter, estimates that 307 000 people experienced homelessness in England in 2017,[3] although this is considered an underestimate as it does not count the vulnerably housed, or those missed by official statistics who often rely on friends and family.

Homelessness is associated with poor health and a high rate of mortality.[4–6] Homeless people often have comorbidities usually found in substantially older people.[1 7] Trimorbidity,[8] is a term used to describe the common combination of physical ill-health, mental ill-health and drug or alcohol misuse which gives rise to complex needs. In 2018, the average age of death for homeless men and women in England and Wales was 45 and 43 years, respectively, compared with 76 and 81 years for the general population.[9] A 2019 study estimated that 30% of deaths among homeless people are from conditions amenable to healthcare, compared with 23% of deaths in a comparison group of housed people from the most deprived quintile of areas in England.[10]

There is some evidence that homeless people have limited access to routine healthcare. Homeless people face barriers to registering with and accessing mainstream primary care services,[11 12] often due to organisational barriers (eg, inflexible appointment schedules) and previous negative experiences, including experience (or fear) of stigma when attempting to access services. People who are homeless are less likely to be registered with a general practitioner than the general population.[13] It is possible that poor access to primary care contributes to the high use of emergency departments and inpatient care by people who are homeless,[7 14 15] which is both costly and lacking in continuity of care for patients who often have complex health problems. The latter is viewed to be especially important for meeting long-term needs and delivering care with compassion for homeless people who have common experiences of trauma.[16]

It has been proposed that emergency hospital admissions related to some conditions, known as ambulatory care-sensitive (ACS) conditions, could be avoided if timely and effective primary care was provided.[17 18] Diabetes, influenza, asthma and anaemia are examples of such conditions. Elevated ACS hospital admissions may also provide indirect insights into the quality of primary care provided, although there are many aspects of quality of care that they will fail to capture.[19] One study has suggested that there were 1.77 million ACS admissions in England during 2011–2012, accounting for 10.9 million bed days.[20]

Currently, there is no study of potentially preventable admissions for homeless patients in England. We add to the evidence using a comparative design: we use national data to match a cohort of homeless patients to a cohort of housed patients with similar observable characteristics. We compare the relative frequency of hospital admissions for each cohort in the following 4 years, and the prevalence of specific ACS conditions. We then discuss the extent to which these admissions may be preventable with improved access to primary care.

## METHODS

In England, administrative data are collected on National Health Service (NHS) hospital service use for all admissions, accident and emergency (A&E) attendances and outpatient appointments.[21] These Hospital Episode Statistics (HES) data contain information on diagnoses and treatment, as well as provider and patient characteristics. Each patient has a unique anonymised identifier which allows individual patients to be tracked when they are treated or seen in different hospital settings and over time.

## Sample definition

We used HES A&E attendances data to identify two groups of patients: housed and homeless. Within HES, there are fields derived from the patient's postcode which assign patients to different administrative geographies.

We used the field for local authority district of residence, where those patients without a postcode are coded 'no fixed abode'.

All patients are registered when they arrive at the emergency department. They are asked for their name, date of birth and address. Whether they are homeless or housed does not affect their entitlement to treatment and there is no direct incentive for patients to not disclose their true status. The emergency department needs the address details to know which health authority to charge for the care episode. They therefore have an incentive to record patients' residential locations accurately as this is needed for billing.

Although there is not a study of the validity of the no fixed abode indicator, the age and sex profile of the patients included in this study who were recorded as having no fixed abode (mean age: 38 years, 76% male) is similar to other studies of people experiencing homelessness in England.[13 22–24]

We defined our sample of homeless patients as those who had an A&E attendance in the 2013/2014 financial year and were recorded as having 'no fixed abode' in the above field. The housed sample were those whose recorded local authority indicated that they resided in England.

For each A&E attendance in 2013/2014, we retained the following variables required to match homeless and housed patients: sex, age, primary diagnosis and the emergency department the patient attended.

We restricted our analysis to patients aged 16–75 years. We removed observations with missing or poor quality data in the fields used for matching, such as if sex was recorded as not specified or not known. Where present, diagnosis codes are a mixture of NHS A&E diagnosis condition codes[25] (92% among observations with 'no fixed abode') and International Classification of Diseases and Related Health Problems (ICD-10) codes.[26]

We retained only the observations with NHS A&E diagnosis codes and matched using only the first two characters, corresponding to 39 broad diagnosis groups such as 'head injury', 'cardiac conditions' and 'respiratory conditions'. We excluded observations where the first two characters were not in the valid range of 01–39 (4% of observations with 'no fixed abode'). These may have been intended to be recorded with a code group between 04 and 09, but were mistakenly coded without the preceding zero. We were unable to distinguish these errors from other coding errors.

Finally, we excluded those patients who had a primary diagnosis for 'social problems' (NHS A&E diagnosis code group 37; 3% of observations with 'no fixed abode') to ensure that homeless patients were identified exclusively on the basis of their area of residence. This prevented homeless patients populating the 'housed' group via this diagnosis code.

We include a flowchart of observations removed at each stage in online supplemental material.

After the above exclusions, the following two mutually exclusive groups of patients remained: (1) 5 345 363 patients recorded as housed in England on all of their A&E attendances during 2013/2014 and (2) 16 338 patients recorded as having no fixed abode on at least one of their A&E attendances during 2013/2014.

## Matching of homeless with housed patients

We used variable ratio matching[27] to match each of the 16 338 homeless patients with up to five housed patients on combinations of sex, single year of age, primary diagnosis and the emergency department the patient attended.

After matching, the two cohorts comprised 16 161 homeless patients and 74 780 housed patients.

## Outcomes

We used each matched patient's unique identifier to locate their admissions in the HES data for the following four financial years (2014/2015, 2015/2016, 2016/2017 and 2017/2018).

We used two outcomes from the Clinical Commissioning Group Outcomes Indicator Set[28] to identify which emergency admissions were for an ACS condition: unplanned hospitalisation for chronic ACS conditions (indicator 2.6) and emergency admission for acute conditions that should not usually require a hospital admission (indicator 3.1).

We constructed five counts for each patient in each financial year: admissions, emergency admissions, emergency admissions for any ACS condition, emergency admissions for an acute ACS condition and emergency admissions for a chronic ACS condition.

If a patient did not appear in the admissions data, we assigned a count of zero admissions for that financial year. Some patients died within the study period, and mortality is likely to differ between the housed and homeless cohorts. We therefore corrected this source of bias using linked death records[29] to remove patients who died from the sample in the years following their death.

Using pooled data for all 4 years, we estimated negative binomial regression models for each outcome due to over-dispersion. We present effects as incident rate ratios (IRRs) and calculated marginal effects for the key variables to show the magnitude of these effects on the original scale. We used weights because each homeless patient is matched to between one and five housed patients. Each homeless patient has a weight of 1 and the weights assigned to housed patients are the reciprocal of the number of housed patients matched to the homeless patient.

For each outcome, we estimated a model which included a binary indicator for whether the patient was recorded as homeless in 2013/2014, year indicators and indicators for the matching characteristics: sex, A&E attended in 2013/2014, primary diagnosis group at A&E attendance in 2013/2014 and 10-year age categories to allow for an expected non-linear relationship between age and all outcomes. We used robust standard errors clustered at the individual level because the sample includes multiple observations for each individual.

## RESULTS

### Matching of homeless with housed patients

We were able to match 16 161 of 16 338 homeless patients to at least one housed patient. Prior to matching, the homeless groups were younger, more likely to be male and were more likely to attend A&E in a London hospital (table 1). While attendances were distributed relatively similarly across the two populations in terms of primary diagnoses, there were a few notable exceptions. First, poisonings (which include overdoses) comprised a higher share of attendances in homeless patients compared with housed: a homeless patient's attendance was for a diagnosis of poisoning 14 times more often. Second, homeless patients' attendances were for psychiatric conditions nine times more often.

Conversely, acute injuries (such as laceration, sprain, dislocation and contusion) were typically more likely for housed patients compared with homeless patients. After matching, the two groups were markedly more balanced in terms of their characteristics.

### Outcomes

The mean values of all outcomes were greater for homeless patients than for housed patients, both before and after matching (table 2).

Each year, most people in the homeless and housed cohorts did not have any admissions. However, for all outcomes, patients in the homeless cohort more frequently had at least one admission each year, in comparison with the housed patients, both before and after matching.

A markedly greater percentage of the homeless cohort had a very large number of admissions and emergency admissions per year, compared with the housed cohort; 0.73% of the homeless cohort had ≥10 admissions each year, compared with 0.12% of the matched housed cohort, and 0.68% of the homeless cohort had ≥10 emergency admissions each year, compared with 0.1% of the housed cohort.

Table 3 presents the estimates from the negative binomial regression models. When the covariates are matched across groups, the homeless cohort had a higher count of all outcomes (table 3; figure 1). For homeless patients, annual admissions were 1.79 times higher (IRR=1.79; 95% CI 1.69 to 1.90), emergency admissions 2.08 times higher (IRR=2.08; 95% CI 1.95 to 2.21) and ACS admissions 1.65 times higher (IRR=1.65; 95% CI 1.51 to 1.80), compared with housed patients. The effect was greater for acute (IRR=1.78; 95% CI 1.64 to 1.93) than chronic (IRR=1.45; 95% CI 1.27 to 1.66) ACS conditions.

The largest marginal effect of homelessness was on emergency admissions; the homeless cohort had 0.225 additional emergency admissions (95% CI 0.198 to 0.252)

**Table 1** Characteristics of homeless and housed groups before and after matching

| | Before matching | | | After matching | | |
|---|---|---|---|---|---|---|
| | **Homeless** | **Housed** | **Difference** | **Homeless** | **Housed** | **Difference** |
| **Individual characteristics** | | | | | | |
| Female (%) | 24 | 49 | −25 | 24 | 24 | 0 |
| Age (years) (SD) | 38.02 (13.66) | 41.45 (16.84) | −3.43 | 37.95 (13.66) | 37.65 (13.66) | 0.3 |
| **Government office region of attendance (%)** | | | | | | |
| East | 13 | 9 | 4 | 13 | 13 | 0 |
| East Midlands | 3 | 8 | −5 | 3 | 3 | 0 |
| London | 31 | 16 | 15 | 31 | 30 | 1 |
| North East | 1 | 4 | −3 | 1 | 1 | 0 |
| North West | 21 | 19 | 2 | 21 | 22 | −1 |
| South East | 16 | 13 | 3 | 16 | 16 | 0 |
| South West | 5 | 9 | −4 | 4 | 4 | 0 |
| West Midlands | 5 | 9 | −4 | 5 | 5 | 0 |
| Yorkshire and the Humber | 5 | 12 | −7 | 5 | 6 | −1 |
| **Primary diagnosis group code (%)\*** | | | | | | |
| Laceration (01) | 5 | 6 | −1 | 5 | 6 | −1 |
| Contusion/abrasion (02) | 3 | 4 | −1 | 3 | 3 | 0 |
| Soft tissue inflammation (03) | 2 | 6 | −4 | 2 | 3 | −1 |
| Head injury (04) | 4 | 3 | 1 | 4 | 5 | −1 |
| Dislocation/fracture/joint injury/ amputation (05) | 4 | 7 | −3 | 4 | 4 | 0 |
| Sprain/ligament injury (06) | 3 | 7 | −4 | 3 | 3 | 0 |
| Muscle/tendon injury (07) | 2 | 3 | −1 | 2 | 2 | 0 |
| Nerve injury (08) | 0.1 | 0.2 | −0.1 | 0.1 | 0.1 | 0 |
| Vascular injury (09) | 0.1 | 0.1 | 0 | 0.1 | 0.03 | 0.07 |
| Burns and scalds (10) | 0.4 | 1 | −0.6 | 0.4 | 0.3 | 0.1 |
| Electric shock (11) | 0.02 | 0.1 | −0.08 | 0.02 | 0.03 | −0.01 |
| Foreign body (12) | 0.5 | 1 | −0.5 | 0.5 | 0.5 | 0 |
| Bites/stings (13) | 0.3 | 1 | −0.7 | 0.3 | 0.3 | 0 |
| Poisoning (inc. overdose) (14) | 14 | 1 | 13 | 14 | 12 | 2 |
| Near drowning (15) | 0.01 | 0.01 | 0 | 0.01 | 0.003 | 0.007 |
| Visceral injury (16) | 0.1 | 0.05 | 0.05 | 0.05 | 0.03 | 0.02 |
| Infectious disease (17) | 1 | 1 | 0 | 1 | 1 | 0 |
| Local infection (18) | 2 | 2 | 0 | 2 | 2 | 0 |
| Septicaemia (19) | 0.1 | 0.2 | −0.1 | 0.1 | 0.05 | 0.05 |
| Cardiac conditions (20) | 2 | 4 | −2 | 2 | 3 | −1 |
| Cerebrovascular conditions (21) | 0.3 | 1 | −0.7 | 0.3 | 0.3 | 0 |
| Other vascular conditions (22) | 0.5 | 1 | −0.5 | 0.5 | 0.3 | 0.2 |
| Haematological conditions (23) | 0.2 | 0.3 | −0.1 | 0.1 | 0.1 | 0.1 |
| Central nervous system conditions (exc. strokes) (24) | 3 | 2 | 1 | 3 | 3 | 0 |
| Respiratory conditions (25) | 2 | 3 | −1 | 2 | 2 | 0 |
| Gastrointestinal conditions (26) | 4 | 6 | −2 | 4 | 5 | −1 |
| Urological conditions (inc. cystitis) (27) | 1 | 2 | −1 | 1 | 1 | 0 |
| Obstetric conditions (28) | 0.2 | 1 | −0.8 | 0.2 | 0.2 | 0 |
| Gynaecological conditions (29) | 0.5 | 2 | −1.5 | 1 | 1 | 0 |

Continued

**Table 1** Continued

| | Before matching | | | After matching | | |
|---|---|---|---|---|---|---|
| | **Homeless** | **Housed** | **Difference** | **Homeless** | **Housed** | **Difference** |
| Diabetes and other endocrinological conditions (30) | 0.3 | 0.4 | –0.1 | 0.3 | 0.2 | 0.1 |
| Dermatological conditions (31) | 0.5 | 1 | –0.5 | 0.5 | 0.4 | 0.1 |
| Allergy (inc. anaphylaxis) (32) | 0.2 | 1 | –0.8 | 0.2 | 0.1 | 0.1 |
| Faciomaxillary conditions (33) | 0.5 | 1 | –0.5 | 0.5 | 0.4 | 0.1 |
| Ear, nose and throat conditions (34) | 1 | 2 | –1 | 1 | 1 | 0 |
| Psychiatric conditions (35) | 9 | 1 | 8 | 9 | 8 | 1 |
| Ophthalmological conditions(36) | 1 | 4 | –3 | 1 | 1 | 0 |
| Diagnosis not classifiable (38) | 26 | 21 | 5 | 26 | 29 | –3 |
| Nothing abnormal detected (39) | 5 | 3 | 2 | 5 | 5 | 0 |
| N | 16 338 | 5 345 363 | | 16 161 | 74 780 | |

Data source: Hospital Episode Statistics A&E 2013/2014; Government office region of attendance displayed instead of A&E attended for brevity.[21]
*NHS accident and emergency codes.

and 0.025 additional ACS admissions (95% CI 0.019 to 0.03) each year compared with the housed cohort.

Admissions for several specific ACS conditions were relatively more common for the homeless cohort than for the housed cohort (table 4). Of the acute ACS conditions, the percentage of total homeless ACS admissions that were for cellulitis was 1.5 times that of the housed cohort (18% compared with 12%), and the percentage for convulsions and epilepsy was 2.4 times that of the housed cohort (12% compared with 5%). Of the chronic ACS conditions, the percentage of homeless ACS admissions for angina was twice that of the housed cohort (8% compared with 4%).

## DISCUSSION
### Summary of main findings
Homeless patients had more admissions in general, more emergency admissions and more admissions for an ACS condition than housed patients with similar characteristics. This effect was greater for acute ACS conditions than for chronic ACS conditions.

We estimated that, compared with 1000 housed patients with similar characteristics, 1000 homeless patients would collectively have 225 more emergency admissions per year and 25 more ACS emergency admissions per year. This suggests that ACS admissions account for around 11% of the additional emergency admissions experienced by homeless patients.

Acute ACS conditions for which emergency admission was more common for the homeless cohort were cellulitis and convulsions/epilepsy, and the one chronic ACS condition was angina.

### Explanation of notable findings
Emergency admission for chronic angina was twice as common in the homeless cohort compared with the housed cohort. This could be due to the relatively high prevalence of cardiovascular disease (CVD) in the homeless population,[6] and/or poor subsequent management of CVD in homeless patients compared with housed patients.

The higher number of acute ACS admissions for homeless patients is driven partly by emergency admissions for cellulitis in particular, and the causes of this are likely to be complex. Hospital admissions related to cellulitis are common among individuals who inject illicit drugs,[30 31] and homelessness is more common among individuals with problematic substance misuse.[22] The differential prevalence of cellulitis in the homeless cohort may therefore be indicative of increased rates of drugs misuse. This may also be tentatively indicated by the higher prevalence of admissions for convulsions, and the higher prevalence of A&E attendances for poisonings in the homeless cohort.

Cellulitis is also strongly associated with comorbidity and conditions such as diabetes; however, the causal relationship is likely to be bidirectional.[30] Furthermore, cellulitis is difficult to diagnose in primary care and even if patients attend primary care, these difficulties may lead to delays in treatment and subsequent and appropriate use of emergency healthcare.[32]

There is likely to be a complex interplay of factors at multiple levels which lead to the differences observed. People experiencing homelessness face barriers to accessing healthcare at the macro-level, such as potential disincentives for providers to provide more accessible services.[15] At the meso-level, organisational factors such as where, how and when appointments are offered in primary care place further barriers to care.[11 12] Finally, at the micro-level, both patient and professional factors are relevant. People experiencing homelessness have been described as having chaotic lifestyles.[15] In combination with the perceived

**Table 2** Descriptive statistics: annual counts of different types of hospital admission for homeless and housed patients

| | n | Mean (SD) | Min | Max | % of observations in range | | | | | | | | |
|---|---|---|---|---|---|---|---|---|---|---|---|---|---|
| | | | | | 0 | 1 | 2 | 3–5 | 6–9 | 10–19 | 20–29 | 30–49 | >50 |
| **Admissions** | | | | | | | | | | | | | |
| Homeless (matched) | 62910 | 0.47 (2.12) | 0 | 127 | 84.06 | 7.68 | 3.22 | 3.24 | 1.07 | 0.57 | 0.09 | 0.04 | 0.03 |
| Housed (matched) | 292842 | 0.23 (0.94) | 0 | 84 | 87.34 | 8.10 | 2.35 | 1.72 | 0.36 | 0.12 | 0.01 | <0.01 | <0.01 |
| Housed (before matching) | 21381452 | 0.23 (0.85) | 0 | 136 | 86.94 | 8.50 | 2.46 | 1.69 | 0.32 | 0.08 | 0.01 | <0.01 | <0.01 |
| **Emergency admissions** | | | | | | | | | | | | | |
| Homeless (matched) | 62910 | 0.43 (2.08) | 0 | 127 | 85.31 | 7.17 | 2.92 | 2.95 | 0.97 | 0.53 | 0.08 | 0.04 | 0.03 |
| Housed (matched) | 292842 | 0.18 (0.83) | 0 | 76 | 89.61 | 6.92 | 1.82 | 1.28 | 0.27 | 0.09 | 0.01 | <0.01 | <0.01 |
| Housed (before matching) | 21381452 | 0.17 (0.71) | 0 | 131 | 89.99 | 6.86 | 1.76 | 1.13 | 0.20 | 0.05 | <0.01 | <0.01 | <0.01 |
| **Emergency admissions for any ACSC** | | | | | | | | | | | | | |
| Homeless (matched) | 62910 | 0.06 (0.65) | 0 | 71 | 96.40 | 2.47 | 0.67 | 0.35 | 0.08 | 0.02 | <0.01 | <0.01 | <0.01 |
| Housed (matched) | 292842 | 0.03 (0.27) | 0 | 19 | 97.56 | 1.91 | 0.33 | 0.16 | 0.03 | 0.01 | 0 | 0 | 0 |
| Housed (before matching) | 21381452 | 0.04 (0.29) | 0 | 44 | 97.26 | 2.14 | 0.39 | 0.19 | 0.03 | 0.01 | <0.01 | <0.01 | 0 |
| **Emergency admissions for acute ACSC** | | | | | | | | | | | | | |
| Homeless (matched) | 62910 | 0.03 (0.25) | 0 | 10 | 97.52 | 1.91 | 0.40 | 0.15 | 0.02 | <0.01 | 0 | 0 | 0 |
| Housed (matched) | 292842 | 0.02 (0.16) | 0 | 9 | 98.49 | 1.32 | 0.14 | 0.04 | <0.01 | 0 | 0 | 0 | 0 |
| Housed (before matching) | 21381452 | 0.02 (0.17) | 0 | 24 | 98.35 | 1.42 | 0.17 | 0.05 | <0.01 | <0.01 | <0.01 | <0.01 | 0 |
| **Emergency admissions for chronic ACSC** | | | | | | | | | | | | | |
| Homeless (matched) | 62910 | 0.03 (0.54) | 0 | 61 | 98.54 | 0.95 | 0.27 | 0.17 | 0.04 | 0.01 | <0.01 | <0.01 | <0.01 |
| Housed (matched) | 292842 | 0.02 (0.21) | 0 | 17 | 98.92 | 0.79 | 0.17 | 0.09 | 0.02 | 0.01 | 0 | 0 | 0 |
| Housed (before matching) | 21381452 | 0.02 (0.22) | 0 | 44 | 98.73 | 0.95 | 0.19 | 0.10 | 0.02 | <0.01 | <0.01 | <0.01 | 0 |

Data source: Hospital Episode Statistics Admitted Patient Care 2014/2015–2017/2018.[21]
ACSC, ambulatory care-sensitive condition.

**Table 3** Negative binomial regression estimates (incident rate ratios) of the effect of homelessness on annual hospital admissions

| | Admissions | | Emergency admissions for an ACSC | | |
|---|---|---|---|---|---|
| | Any (1) | Emergency (2) | Any ACSC (3) | Acute (4) | Chronic (5) |
| Homelessness | 1.79*** | 2.08*** | 1.65*** | 1.78*** | 1.45*** |
| | (1.69 to 1.90) | (1.95 to 2.21) | (1.51 to 1.80) | (1.64 to 1.93) | (1.27 to 1.66) |
| Year of admission | | | | | |
| 2014 (reference category) | | | | | |
| 2015 | 0.82*** | 0.82*** | 0.87*** | 0.87** | 0.86** |
| | (0.79 to 0.84) | (0.79 to 0.85) | (0.81 to 0.93) | (0.80 to 0.95) | (0.77 to 0.96) |
| 2016 | 0.71*** | 0.72*** | 0.87*** | 0.88** | 0.82*** |
| | (0.69 to 0.74) | (0.69 to 0.75) | (0.80 to 0.93) | (0.81 to 0.97) | (0.74 to 0.92) |
| 2017 | 0.68*** | 0.70*** | 0.84*** | 0.82*** | 0.84** |
| | (0.65 to 0.71) | (0.67 to 0.73) | (0.77 to 0.91) | (0.75 to 0.91) | (0.74 to 0.95) |
| Sex | | | | | |
| Female | 1.32*** | 1.11** | 1.14* | 1.21*** | 1.00 |
| | (1.24 to 1.40) | (1.04 to 1.19) | (1.03 to 1.26) | (1.10 to 1.34) | (0.85 to 1.19) |
| Age at time of attendance (years) | | | | | |
| 16–25 (reference category) | | | | | |
| 26–35 | 1.19*** | 1.24*** | 1.40*** | 1.60*** | 1.11 |
| | (1.09 to 1.29) | (1.13 to 1.36) | (1.21 to 1.63) | (1.39 to 1.84) | (0.84 to 1.46) |
| 36–45 | 1.57*** | 1.71*** | 1.98*** | 2.18*** | 1.74*** |
| | (1.43 to 1.72) | (1.55 to 1.88) | (1.71 to 2.28) | (1.90 to 2.49) | (1.37 to 2.22) |
| 46–55 | 1.87*** | 1.99*** | 2.48*** | 2.18*** | 3.06*** |
| | (1.70 to 2.06) | (1.80 to 2.21) | (2.14 to 2.89) | (1.89 to 2.53) | (2.40 to 3.91) |
| 56–65 | 1.99*** | 2.07*** | 3.31*** | 2.31*** | 5.07*** |
| | (1.79 to 2.21) | (1.84 to 2.32) | (2.81 to 3.91) | (1.96 to 2.72) | (3.90 to 6.60) |
| 66–75 | 3.17*** | 3.49*** | 7.30*** | 4.73*** | 11.34*** |
| | (2.42 to 4.15) | (2.64 to 4.63) | (4.69 to 11.37) | (3.74 to 5.99) | (6.33 to 20.34) |
| n | 355 752 | 355 752 | 355 752 | 355 752 | 355 752 |

Data source: Hospital Episode Statistics Admitted Patient Care 2014/2015–2017/2018.[21]
*p<0.05; **p<0.01; ***p<0.001.
ACSC, ambulatory care-sensitive condition.

and/or experienced stigma and discrimination from health professionals, this may lead to poor or delayed engagement with the health service. These social and structural issues mean that health problems can degenerate into more serious problems that subsequently require contact with emergency services.

### Comparison with the existing research

A study of homeless patients admitted to hospital in California in 2010 found that the odds that the admission was ACS was increased when the patient was black, admitted from the emergency department or transferred from another healthcare facility, or did not have Medicare.[33] A study of hospital inpatient records in Ireland from 2005 to 2014 found that

of 2051 emergency admissions for patients with 'no fixed abode', 280 (13.7%) were for an ACS condition. The most common ACS conditions among the admissions were convulsions/epilepsy (32.9%), cellulitis (22.1%) and chronic obstructive pulmonary disease (10.4%).[34] Neither study compared ACS admissions of homeless patients with those of housed patients.

We add to the current evidence using a comparative design: we matched a group of homeless patients to a group of housed patients with similar observable characteristics. We also used more recent data so our analysis is reflective of factors which have changed in the 2010s such as economic policies enacted by the

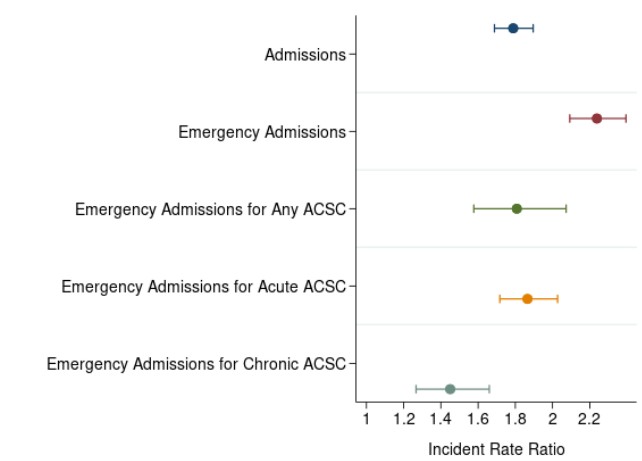

**Figure 1** The effect of homelessness on outcomes with 95% CIs. Data source: HES APC 2014/2015–2017/2018; full regression results in table 3. ACSC, ambulatory care-sensitive condition.

UK government and the increase in homelessness in England.[23]

## Limitations

The no fixed abode flag in HES is not a perfect indicator of homelessness. Some patients are recorded as having no fixed abode on one or more attendances and as being homed on other attendances. In part, this will be the result of genuine changes in housing status, but we also suspect that (although there is no direct incentive for them to do so) false information may be provided in some cases, such as the address of a friend or relative. This may bias our estimates towards the null.

Several factors may affect the generalisability of our study. First, it is possible that patients recorded as having no fixed abode are some of the most vulnerable and isolated homeless people who do not have an address to give. Second, because there is no comprehensive database of all people experiencing homelessness, we selected patients to include in the study on the basis of A&E attendances. These patients may be more likely to be admitted in an emergency than less regular A&E attendees, and the size of this difference may vary across the two cohorts.

We used linked death records to remove patients who died from the sample in the years following the year in which they died. The mortality rates were 5.06% in the homeless cohort and 3.76% in the housed cohort within the study period. We included the count of admissions during a financial year when the patient died part way through the financial year. This biases the difference in admission rates between the homeless and housed cohorts downwards due to differences in exposure time.

We matched homeless patients to housed patients on the basis of sex, age, emergency department attended in 2013/2014, and the diagnosis received

**Table 4** Count and percentage of ambulatory care-sensitive (ACS) emergency admissions for specific conditions

|  | Housed (% of total) | Homeless (% of total) |
| --- | --- | --- |
| **Acute** |  |  |
| Influenza, pneumonia, other vaccine preventable | 829 (8%) | 259 (6%) |
| Angina | 108 (1%) | 57 (1%) |
| Dehydration and gastroenteritis | 833 (8%) | 193 (5%) |
| Pyelonephritis and kidney/urinary tract infections | 958 (10%) | 180 (5%) |
| Perforated/bleeding ulcer | 290 (3%) | 82 (2%) |
| Cellulitis | 1148 (12%) | 722 (18%) |
| Ear, nose and throat | 331 (4%) | 50 (1%) |
| Dental | 188 (2%) | 71 (2%) |
| Convulsions and epilepsy | 492 (5%) | 497 (12%) |
| **Chronic** |  |  |
| Chronic viral hepatitis B | 2 (<1%) | 0 (0%) |
| Diabetes | 773 (8%) | 322 (8%) |
| Anaemia | 114 (1%) | 38 (1%) |
| Epilepsy/dementia/atrial | 1410 (14%) | 587 (14%) |
| Heart failure | 302 (3%) | 41 (1%) |
| Angina | 378 (4%) | 335 (8%) |
| Hypertension | 57 (1%) | 8 (<1%) |
| Bronchitis | 1109 (11%) | 402 (10%) |
| Asthma | 643 (6%) | 156 (4%) |
| Total count of ACS emergency admissions | 9965 | 4000 |
| Number of patients | 74 780 | 16 161 |

Data source: Hospital Episode Statistics Admitted Patient Care 2014/15-2017/18; percentages are rounded to the nearest whole number and may not add to 100.[21]
ACS, ambulatory care-sensitive.

on that attendance. Matching on this combination of variables allowed us to match 16 161 homeless patients (of the total possible 16 338) to at least one housed patient. However, as with all matching studies, there is a threat of residual confounding.

## Implications for policy and research

Dixon-Woods et al[35] have previously drawn attention to the complex interplay of factors (perceptions of need, priorities, attitudes of staff, resources and policies) that create inequalities and vulnerabilities for marginalised and disadvantaged groups in terms of access to healthcare. Additional research has highlighted this complex interplay of barriers regarding access to primary care that can be viewed as creating a two-tier system for marginalised or 'hard-to-reach' groups.[36 37]

Our results suggest that patients who have experienced homelessness are admitted to hospital more frequently than housed patients for conditions classified as suitable for treatment and management in a primary care setting. Addressing structural factors such as improved access to primary care could potentially reduce some of these admissions. However, the fact that only 11% of the additional emergency admissions experienced by the homeless cohort were ACS suggests that other approaches are needed if the goal is to reduce homeless persons' admissions to hospital. Our results may suggest that interventions which strengthen support for homeless A&E attendees have the potential to reduce future hospital admissions.

Additionally, the likely relevance of other micro-level factors as a driver of some of the ACS admissions is a reminder of the complexity of health problems experienced by people who are homeless, which often have causes that are outside the control of the healthcare system. The link between ACS admissions and access to primary care is less clear for this population than it might be for others.

**Author affiliations**
[1]Health Organisation, Policy and Economics, Centre for Primary Care and Health Services Research, School of Health Sciences, The University of Manchester, Manchester, UK
[2]Melbourne Institute: Applied Economic and Social Research, The University of Melbourne, Melbourne, Victoria, Australia
[3]NIHR Greater Manchester Patient Safety Translational Research Centre, Centre for Primary Care and Health Services Research, Manchester Academic Health Science Centre, The University of Manchester, Manchester, UK
[4]Manchester Centre for Health Economics, School of Health Sciences, The University of Manchester, Manchester, UK
[5]Danish Centre for Health Economics, Department of Public Health, University of Southern Denmark, Odense, Denmark

**Contributors** SC-S and CS conceived the original idea. MS and TA designed the empirical approach. CM performed data analysis and drafted the first version of the manuscript. All authors contributed to interpretation of results and editing the manuscript for the final version.

**Funding** This research was funded jointly by a NIHR Research Methods Fellowship (RM-FI-2017-08-018) and the NIHR Greater Manchester Patient Safety Translational Research Centre (PSTRC-2016-003).

**Disclaimer** The views expressed are those of the authors and not necessarily those of the NIHR or the Department of Health and Social Care.

**Competing interests** None declared.

**Patient consent for publication** Not required.

**Provenance and peer review** Not commissioned; externally peer reviewed.

**Data availability statement** The Hospital Episode Statistics data that support the findings of this study are available from NHS Digital. Restrictions apply to the availability of these data. They were used under license for the current study, and so are not publicly available.

**ORCID iDs**
Charlie Moss http://orcid.org/0000-0002-4694-378X
Thomas Allen http://orcid.org/0000-0002-2972-7911

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
