## [Reviewer comments · BMJ Open]

ARTICLE DETAILS

TITLE (PROVISIONAL)	Comparative four year risk and type of hospital admission among homeless and housed emergency department attendees: longitudinal study of hospital records in England 2013-2018
AUTHORS	Moss, Charlie; Sutton, Matt; Cheraghi-Sohi, Sudeh; Sanders, Caroline; Allen, Thomas

VERSION 1 – REVIEW

REVIEWER	Jacques, Richard University of Sheffield, School of Health and Related Research
REVIEW RETURNED	23-Feb-2021

GENERAL COMMENTS	This paper reports the results of a retrospective matched cohort study investigating the number and type of hospital admissions among homeless compared to housed emergency department attendees using Hospital Episode Statistics for the financial years 2013/14 to 2017/18. HES A&E attendances from 2013/14 were used to identify the two groups of patients (housed and homeless). The homeless were identified using the code for “no fixed abode” in the local authority district of residence field. I do not have any experience of using this field to identify homeless attendees so I can’t comment on the completeness or appropriateness but the authors have commented the limitations of its use. The other limitation would be the generalisability of the results as both groups are selected from people attending the emergency department so they may be more likely to attend or be admitted in the future. This is however acknowledged in the discussion. The majority of the methods and results are clearly described and the majority of my comments are clarifications or suggestions for improved reporting. 1) A&E Primary Diagnosis is used in the matching of homeless and housed patients. My experience of this field is that the data are poorly recorded and there are potentially ambiguous codes (when proceeding zeroes are missing). On page 8 it says that observations were removed due to poor data quality is this what you are referring to? I would suggest making that clear. 2) Page 9, paragraph 2: “Some patients died within the study period, and mortality is likely to differ between the house and homeless cohorts. We therefore corrected this source of bias by using linked death records to remove patients who die from the sample in the years following their death”. It is good that you were
---

	able to use linked death records to remove patients who died. You say that patients who die are removed from the sample in the years following their death but what happens in the year that they died? For example if some dies part way through a financial year are the admissions for that year included? If so, wouldn't this also potentially create a bias due to differences in exposure time? 3) Page 9, paragraphs 4 and 5: I have a number of questions/comments about the models that you have used: a. "Negative binomial regression models were used for each outcome". Were negative binomial models used over Poisson models due to the outcomes being over-dispersed? b. The regression models have been weighted to allow for each homeless patient being matched to a different number of housed patients (between 1 and 5). Although weighting the controls sounds like a sensible approach it is not something that I am familiar with. I think including a reference to this approach would be good (if there is one). c. There are multiple counts per individual in the model (potentially one for each of the four years). This means that the observations are not independent. I assume this is why you have used robust standard errors but I think this should be made clear in the methods. 4) Table 1: I found the labelling of the table a bit confusing. The columns are labelled "No fixed abode mean" and "Homed mean" but show both mean and percentages (the majority being percentages). I would remove the mean label and then make it clear that only age is a mean. I also wouldn't normally report standard deviations for proportions/percentages because they can be directly calculated. 5) Table 3: I would prefer to see 95% confidence intervals for the incident rate ratios presented here rather than the standard errors.
--	--

REVIEWER	Miyawaki, Atsushi The University of Tokyo, Department of Public Health, Graduate School of Medicine
REVIEW RETURNED	04-Mar-2021

GENERAL COMMENTS	This paper investigated how different the no. of admissions were between homeless patients who visited ED at least once and housed patients who visited ED at least once across England in 2013/2014. The results were consistent to several studies in Canada and US-- homeless patients had higher risk of admissions. They further focused on ambulatory care sensitive conditions and found homeless patients had higher risk of ACS admissions. The paper is well written, and the results are understandable, but I have several concerns about the methodology of this paper. Major 1. This study used whether the patients has no fixed adobe in the hospital episode statistics (HES) data as an indicator of homelessness. Is this definition valid? Is there a validity study? Or is it correct to say that, procedurally, this variable is very likely to reflect true homeless status? I read the manuscript carefully, but
---

	was unable to obtain any of this information (slightly discussed in limitation). I am not familiar with this dataset in England (I am familiar with studies related to homeless in Japan and the U.S.). Other international readers would be the same. So any information on the details of this homeless definition would be helpful. This variable is an an important exposure in this study, and the justification for its use should be reinforced. 2. This study targeted on only patients who visited emergency department at once in FY 2013/2014. The pattern of visits to the emergency department (ED) would be different between homeless and non-homeless people. Homeless people may be more likely to visit the ED because they have less access to primary care. The homeless may also visit the ED to secure a bed, food, and social services in addition to treating their illness. Therefore, it is not surprising that such homeless people have a higher rate of hospitalization (even after adjusted for several confounders). However, it is not clear from this study whether homeless people are more likely to be hospitalized than non-homeless people in the general population (including those who do not visit EDs). This issue of sampling bias should be discussed further (eg limitation). Also, authors can re-think the meaning of this study. For example, the results of this study may justify some interventions for the homeless, especially in the ED, even if the results are only true for patients seen in the ED. Minor 1. Be consistent about the expression of "homeless." e.g., in page 10 line 42-44 ,there is mixed expressions like "people with no fixed adobe" and "homeless patients." Consistency is needed throughout the manuscript (I prefer to "homeless").
--	---

REVIEWER	Tsugawa, Yusuke David Geffen School of Medicine at UCLA, Medicine
REVIEW RETURNED	05-Mar-2021

GENERAL COMMENTS	Thank you for giving me an opportunity to review the manuscript by Moss et al. Using the hospital record data in England from 2013 through 2018, authors reported that homeless individuals experienced higher rates of admissions and emergency admissions compared with housed individuals. They also found that the differences were larger for acute compared with chronic ACS conditions. While I think the manuscript is well-written, there are several issues that need to be addressed. First, I am concerned about the confounding. In particular, I think the control group (housed individuals) used in this study is not comparable with housed individuals, making it difficult to interpret whether the observed differences in outcome variables are due to homelessness or residual confounding (i.e., homeless individuals were sicker and thus required more healthcare services). Given that 16,338 homeless patients were matched to the control selected from 5,345,363 housed patients, I believe that more precise matching should be possible. For example, instead of matching on 39 broad diagnosis groups, I suggest authors match on more granular diagnosis (exact ICD-10 codes of the primary diagnosis). Also, instead of matching on 109 CCG region of the emergency department the patient attended, I suggest authors
--

	match on the indicator of the exact emergency department (i.e., emergency department fixed effects), which leads to less bias. Second, the lack of data on homelessness is the major limitation of this study. The authors assumed that patients with “no fixed abode” be homeless. However, it is likely that a large number of homeless patients report the address of their friend’s house or homeless shelter. Given that these patients were coded as housed in their data, it is important to note that this might only bias their estimates towards the null. That said, it is an important limitation to emphasize in this paper. Finally, if multiple observations could be included in the data, a more robust study design may be to adjust for patient fixed effects. This is sometimes called a “self-controlled case series” design. This is a more robust causal design because instead of comparing different individuals, this study design allows authors to compare the same individuals under different situations: homeless vs. housed. I am not sure if the authors have enough sample size to do this analysis, but I would like to see how results change by this specification, at least as a sensitivity analysis.
--	---

VERSION 1 – AUTHOR RESPONSE

Reviewer 1

Dr. Richard Jacques, University of Sheffield

Comments to the Author:

This paper reports the results of a retrospective matched cohort study investigating the number and type of hospital admissions among homeless compared to housed emergency department attendees using Hospital Episode Statistics for the financial years 2013/14 to 2017/18.

HES A&E attendances from 2013/14 were used to identify the two groups of patients (housed and homeless). The homeless were identified using the code for “no fixed abode” in the local authority district of residence field. I do not have any experience of using this field to identify homeless attendees so I can’t comment on the completeness or appropriateness but the authors have commented the limitations of its use.

The other limitation would be the generalisability of the results as both groups are selected from people attending the emergency department so they may be more likely to attend or be admitted in the future. This is however acknowledged in the discussion.

The majority of the methods and results are clearly described and the majority of my comments are clarifications or suggestions for improved reporting.

1) A&E Primary Diagnosis is used in the matching of homeless and housed patients. My experience of this field is that the data are poorly recorded and there are potentially ambiguous codes (when proceeding zeroes are missing). On page 8 it says that observations were removed due to poor data quality is this what you are referring to? I would suggest making that clear.

We have provided further clarification (in Methods, sample definition subsection, paragraphs 5 and 6): “We restricted our analysis to patients aged 16 to 75. We removed observations with missing or poor quality data in the fields used for matching, such as if sex was recorded as not specified or not known. Where present, diagnosis codes are a mixture of NHS A&E diagnosis condition codes²² (92% amongst observations with “no fixed abode”) and ICD-10 codes.²³

We retained only the observations with NHS A&E diagnosis codes and matched using only the first two characters, corresponding to 39 broad diagnosis groups such as “head injury”, “cardiac conditions”, and “respiratory conditions”. We excluded observations where the first two characters

were not in the valid range of 01 to 39 (4% of observations with “no fixed abode”). These may have been intended to be recorded with a code group between 04 and 09, but were mistakenly coded without the preceding zero. We were unable to distinguish these errors from other coding errors.”

2) Page 9, paragraph 2: “Some patients died within the study period, and mortality is likely to differ between the house and homeless cohorts. We therefore corrected this source of bias by using linked death records to remove patients who die from the sample in the years following their death”. It is good that you were able to use linked death records to remove patients who died. You say that patients who die are removed from the sample in the years following their death but what happens in the year that they died? For example if some dies part way through a financial year are the admissions for that year included? If so, wouldn't this also potentially create a bias due to differences in exposure time?

We included the admissions of patients who died within the financial year. The reviewer is correct that this causes a downward bias in the difference between the homeless and housed cohorts because the homeless cohort have a higher mortality rate. The mortality rates were 5.06% in the homeless cohort and 3.76% in the housed cohort within the study period. The bias is important but small in magnitude relative to the difference in admission rates.

We have now included this in the limitations section as follows:

“We used linked death records to remove patients who died from the sample in the years following the year in which they died. The mortality rates were 5.06% in the homeless cohort and 3.76% in the housed cohort within the study period. We included the count of admissions during a financial year when the patient died part way through the financial year. This biases the difference in admission rates between homeless and housed cohorts downwards due to differences in exposure time.”

3) Page 9, paragraphs 4 and 5: I have a number of questions/comments about the models that you have used:

a. “Negative binomial regression models were used for each outcome”. Were negative binomial models used over Poisson models due to the outcomes being over-dispersed?

We used negative binomial models due to over-dispersion, and have clarified this (Methods, outcomes subsection, paragraph 5):

“Using pooled data for all four years, we estimated negative binomial regression models for each outcome due to over-dispersion. We present effects as Incident Rate Ratios (IRRs) and calculated marginal effects for the key variables to show the magnitude of these effects on the original scale. We used weights because each homeless patient is matched to between one and five housed patients. Each homeless patient has a weight of 1 and the weights assigned to housed patients are the reciprocal of the number of housed patients matched to the homeless patient.”

b. The regression models have been weighted to allow for each homeless patient being matched to a different number of housed patients (between 1 and 5). Although weighting the controls sounds like a sensible approach it is not something that I am familiar with. I think including a reference to this approach would be good (if there is one).

We have added the term “variable ratio matching” for the approach we have taken with a reference to Stuart (2010) who describes this approach (Methods, matching subsection, paragraph 1):

“We used variable ratio matching²⁷ to match each of the 16,338 homeless patients with up to five housed patients on combinations of sex, single year of age, primary diagnosis and the emergency department the patient attended.”

c. There are multiple counts per individual in the model (potentially one for each of the four years). This means that the observations are not independent. I assume this is why you have used robust standard errors but I think this should be made clear in the methods.

We have now clarified that this is why we used robust standard errors (Methods, outcomes subsection, paragraph 6):

“For each outcome, we estimated a model which included a binary indicator for whether the patient

was recorded as homeless in 2013/14, year indicators, and indicators for the matching characteristics: sex, A&E attended in 2013/14, primary diagnosis group at A&E attendance in 2013/14, and ten-year age categories to allow for an expected non-linear relationship between age and all outcomes. We used robust standard errors clustered at the individual level because the sample includes multiple observations for each individual.”

4) Table 1: I found the labelling of the table a bit confusing. The columns are labelled “No fixed abode mean” and “Homed mean” but show both mean and percentages (the majority being percentages). I would remove the mean label and then make it clear that only age is a mean. I also wouldn’t normally report standard deviations for proportions/percentages because they can be directly calculated. We have changed Table 1 to remove the “mean” part of the main headings. Instead we have now put mean in brackets next to age, and % in brackets next to the other subheadings. We have also removed the standard deviations for the percentages.

5) Table 3: I would prefer to see 95% confidence intervals for the incident rate ratios presented here rather than the standard errors.

We have changed Table 3 to show confidence intervals instead of standard errors

Reviewer: 2

Dr. Atsushi Miyawaki, The University of Tokyo

Comments to the Author:

This paper investigated how different the no. of admissions were between homeless patients who visited ED at least once and housed patients who visited ED at least once across England in 2013/2014. The results were consistent to several studies in Canada and US-- homeless patients had higher risk of admissions. They further focused on ambulatory care sensitive conditions and found homeless patients had higher risk of ACS admissions. The paper is well written, and the results are understandable, but I have several concerns about the methodology of this paper.

Major

1. This study used whether the patients has no fixed abode in the hospital episode statistics (HES) data as an indicator of homelessness. Is this definition valid? Is there a validity study? Or is it correct to say that, procedurally, this variable is very likely to reflect true homeless status? I read the manuscript carefully, but was unable to obtain any of this information (slightly discussed in limitation). I am not familiar with this dataset in England (I am familiar with studies related to homeless in Japan and the U.S.). Other international readers would be the same. So any information on the details of this homeless definition would be helpful. This variable is an an important exposure in this study, and the justification for its use should be reinforced.

We have amended the manuscript to include more information on the procedure by which people are recorded as having no fixed abode. Although there is no study of the validity of the no fixed abode indicator, the age and sex profile are comparable to other studies of people experiencing homelessness in England. We have added 2 paragraphs with this information (Methods section, sample definition subsection, paragraphs 2 and 3):

“All patients are registered when they arrive at the emergency department. They are asked for their name, date of birth and address. Whether they are homeless or housed does not affect their entitlement to treatment and there is no direct incentive for patients to not disclose their true status. The emergency department needs the address details to know which health authority to charge for the care episode. They therefore have an incentive to record patients’ residential locations accurately as this is needed for billing.

Although there is not a study of the validity of the no fixed abode indicator, the age and sex profile of the patients included in this study who were recorded as having no fixed abode (mean age 38, 76% male) is similar to other studies of people experiencing homelessness in England.^{13,22–24”}

2. This study targeted on only patients who visited emergency department at once in FY 2013/2014.

The pattern of visits to the emergency department (ED) would be different between homeless and non-homeless people. Homeless people may be more likely to visit the ED because they have less access to primary care. The homeless may also visit the ED to secure a bed, food, and social services in addition to treating their illness. Therefore, it is not surprising that such homeless people have a higher rate of hospitalization (even after adjusted for several confounders). However, it is not clear from this study whether homeless people are more likely to be hospitalized than non-homeless people in the general population (including those who do not visit EDs).

This issue of sampling bias should be discussed further (eg limitation). Also, authors can re-think the meaning of this study. For example, the results of this study may justify some interventions for the homeless, especially in the ED, even if the results are only true for patients seen in the ED.

The reviewer is correct to highlight that we are comparing two cohorts of people who have visited an emergency department, not the full population of homeless people for whom there is no comprehensive database. We noted the limitations that this placed on generalisability in the original limitations section of the Discussion (paragraph 2) but we have now edited this to give it more emphasis:

“Several factors may affect the generalisability of our study. First, it is possible that patients recorded as having no fixed abode are some of the most vulnerable and isolated homeless people who do not have an address to give. Second, because there is no comprehensive database of all people experiencing homelessness, we selected patients to include in the study on the basis of A&E attendances. These patients may be more likely to be admitted in an emergency than less regular A&E attendees, and the size of this difference may vary across the two cohorts.”

We have also added the suggestion that our results may suggest the strengthening of support and interventions for homeless people in emergency departments (Discussion, implications for policy and research):

“Our results suggest that patients who have experienced homelessness are admitted to hospital more frequently than housed patients for conditions classified as suitable for treatment and management in a primary care setting. Addressing structural factors such as improved access to primary care could potentially reduce some of these admissions. However, the fact that only 11% of the additional emergency admissions experienced by the homeless cohort were ACS suggests that other approaches are needed if the goal is to reduce homeless persons’ admissions to hospital. Our results may suggest that interventions which strengthen support for homeless A&E attendees have the potential to reduce future hospital admissions.”

Minor

Be consistent about the expression of "homeless." e.g., in page 10 line 42-44 ,there is mixed expressions like "people with no fixed adobe" and "homeless patients." Consistency is needed throughout the manuscript (I prefer to "homeless").

We have changed all incidences to “homeless patients”.

Reviewer: 3

Dr. Yusuke Tsugawa, David Geffen School of Medicine at UCLA

Comments to the Author:

Thank you for giving me an opportunity to review the manuscript by Moss et al. Using the hospital record data in England from 2013 through 2018, authors reported that homeless individuals experienced higher rates of admissions and emergency admissions compared with housed individuals. They also found that the differences were larger for acute compared with chronic ACS conditions. While I think the manuscript is well-written, there are several issues that need to be addressed.

First, I am concerned about the confounding. In particular, I think the control group (housed

individuals) used in this study is not comparable with housed individuals, making it difficult to interpret whether the observed differences in outcome variables are due to homelessness or residual confounding (i.e., homeless individuals were sicker and thus required more healthcare services). Given that 16,338 homeless patients were matched to the control selected from 5,345,363 housed patients, I believe that more precise matching should be possible. For example, instead of matching on 39 broad diagnosis groups, I suggest authors match on more granular diagnosis (exact ICD-10 codes of the primary diagnosis). Also, instead of matching on 109 CCG region of the emergency department the patient attended, I suggest authors match on the indicator of the exact emergency department (i.e., emergency department fixed effects), which leads to less bias.

Unfortunately the more detailed ICD-10 codes aren't available in the emergency department data for England. Instead, they record diagnoses using NHS A&E diagnosis codes, which are consistently recorded into only 39 broad groups.

Following the reviewer's suggestion, we have now altered our approach to match on the emergency department attended instead of region of attendance. This leaves us with a very similar sample size and slightly reduces the magnitude of the differences between the matched homeless and housed cohorts.

We added to one of the limitations to explicitly say that there is a threat of residual confounding (limitations, paragraph 4):

"We matched homeless patients to housed patients on the basis of sex, age, emergency department attended in 2013/14, and the diagnosis received on that attendance. Matching on this combination of variables allowed us to match 16,161 homeless patients (out of the total possible 16,338) to at least one housed patient. However, as with all matching studies, there is a threat of residual confounding."

Second, the lack of data on homelessness is the major limitation of this study. The authors assumed that patients with "no fixed abode" be homeless. However, it is likely that a large number of homeless patients report the address of their friend's house or homeless shelter. Given that these patients were coded as housed in their data, it is important to note that this might only bias their estimates towards the null. That said, it is an important limitation to emphasize in this paper.

We also suspect that some homeless patients do give an address so are not coded as having no fixed abode. However it is likely that patients who aren't homeless will get coded as no fixed abode much less frequently. As you say, this is likely to bias our estimate towards the null.

We have added a sentence on this (Limitations subsection in discussion, paragraph 1):

"The no fixed abode flag in HES is not a perfect indicator of homelessness. Some patients are recorded as having no fixed abode on one or more attendances and as being homed on other attendances. In part this will be the result of genuine changes in housing status, but we also suspect that false information may be provided in some cases, such as the address of a friend or relative. This may bias our estimates towards the null."

Additionally, in response to this comment and reviewer 2's first comment, we amended the manuscript to add 2 paragraphs to clarify the procedure by which patients are recorded as having no fixed abode, and the validity of the no fixed abode indicator (Methods section, sample definition subsection, paragraphs 2 and 3; restated below). Although some homeless people may give the address of a friend or relative, there is no direct incentive for them to do this in terms of receiving health care:

"All patients are registered when they arrive at the emergency department. They are asked for their name, date of birth and address. Whether they are homeless or housed does not affect their entitlement to treatment and there is no direct incentive for patients to not disclose their true status. The emergency department needs the address details to know which health authority to charge for the care episode. They therefore have an incentive to record patients' residential locations accurately as this is needed for billing.

Although there is not a study of the validity of the no fixed abode indicator, the age and sex profile of the patients included in this study who were recorded as having no fixed abode (mean age 38, 76%

male) is similar to other studies of people experiencing homelessness in England.13,22–24”

Finally, if multiple observations could be included in the data, a more robust study design may be to adjust for patient fixed effects. This is sometimes called a “self-controlled case series” design. This is a more robust causal design because instead of comparing different individuals, this study design allows authors to compare the same individuals under different situations: homeless vs. housed. I am not sure if the authors have enough sample size to do this analysis, but I would like to see how results change by this specification, at least as a sensitivity analysis.

We used the study design we have done because it is less problematic to take a snapshot of patients who experienced homelessness in a certain year and compare them to patients who were not recorded as experiencing homelessness at all in the same year. We look at a population that has experienced homelessness, rather than a population who are necessarily currently in the homeless state.

A self-controlled case series design would require us to accurately assign periods of time as homeless and housed for each patient. We only observe patients’ residential status when they attend hospital and we would therefore not be able to define exposure periods accurately. When a patient has a hospital interaction, and is recorded as homeless, they could have been homeless for a period of time prior to that interaction. For this reason we can’t use the emergency department visit dates to define time of exposure to homelessness.

Moreover, a self-controlled case series design would require us to assume when homelessness affects an individual’s risk of hospitalisation. This is unlikely to be instantaneous or instantaneously reversed when the patient is re-housed. For this reason, we examine the consequences of experiencing homelessness in the 4 years after homelessness is recorded.

VERSION 2 – REVIEW

REVIEWER	Jacques, Richard University of Sheffield, School of Health and Related Research
REVIEW RETURNED	14-Jun-2021

GENERAL COMMENTS	The authors have addressed all of the comments from my original review. I have no further comments and I am happy to recommend that the paper is accepted. Thank you for the opportunity to read and review this interesting paper.
---

REVIEWER	Miyawaki, Atsushi The University of Tokyo, Department of Public Health, Graduate School of Medicine
REVIEW RETURNED	24-Jun-2021

GENERAL COMMENTS	I think the authors addressed all the reviewers' concerns in an appropriate manner. I do not have further comments.
---